# DOMAIN ADAPTATION VIA LOW-RANK BASIS APPROXIMATION

## ABSTRACT

Domain adaptation focuses on the reuse of supervised learning models in a new context. Prominent applications can be found in robotics, image processing or web mining. In these areas, learning scenarios change by nature, but often remain related and motivate the reuse of existing supervised models. While the majority of symmetric and asymmetric domain adaptation algorithms utilize all available source and target domain data, we show that efficient domain adaptation requires only a substantially smaller subset from both domains. This makes it more suitable for real-world scenarios where target domain data is rare. The presented approach finds a target subspace representation for source and target data to address domain differences by orthogonal basis transfer. By employing a low-rank approximation, the approach remains low in computational time. The presented idea is evaluated in typical domain adaptation tasks with standard benchmark data.

## 1 INTRODUCTION

Supervised learning and in particular classification is an essential task in machine learning with a broad range of applications. The obtained models are used to predict the label of unseen test samples. A basic assumption in supervised learning is that the underlying domain or distribution is not changing between training and test samples. If the domain is changing from one task to a related but different task, one would like to reuse the available learning model. Domain differences are quite common in real-world scenarios and eventually, lead to substantial performance drops (Weiss et al., 2016).

A domain adaptation example is the classification of web pages. A classifier is trained in the domain of university web pages with a word distribution according to universities. In the test scenario, the domain has changed to non-university web pages, where the word distribution is related, but may not be similar to the training distribution. More formally, let $\mathbf{Z} = \{\mathbf{z}_1, \ldots, \mathbf{z}_m\} \in \mathbb{R}^d$ be source data sampled from the source domain distribution $p(z)$ and let $\mathbf{X} = \{\mathbf{x}_1, \ldots, \mathbf{x}_n\} \in \mathbb{R}^d$ be target data from the target domain distribution $p(x)$. Traditional machine learning assumes similar distributions, i.e. $p(z) \sim p(x)$, but domain adaptation assumes different distributions, i.e. $p(z) \neq p(x)$ and appears in the web page example where $\mathbf{Z}$ could be data of university websites and $\mathbf{X}$ are data of non-university websites.

Multiple domain adaptation techniques have been already proposed, following different strategies and improving the prediction performance of underlying classification algorithms in test scenarios (Weiss et al., 2016; Pan & Yang, 2010). The common domain adaptation approach requires a large number of source or target samples, which is indeed a disadvantage of many domain adaptation approaches and is not guaranteed in restricted environments where labeling is expensive (Weiss et al., 2016). Further, learning a matrix for the alignment of domain distributions does not necessarily lead to effective adaptation shown by (Raab & Schleif, 2018), where domain differences are explicitly solved by overriding the basis of one domain with another and is competitive to current state of the art domain adaptation algorithms (Gong et al., 2012; Long et al., 2015; 2013; Pan et al., 2011; Fernando et al., 2013; Sun et al., 2016).

The main contribution of this work is to improve the Basis-Transfer (Raab & Schleif, 2018) approach by deriving a more simplified closed-form solution of the least-squares problem and enhance this by a Nyström based dimensionality reduction. The resulting method has a better prediction performance and makes it easer to apply. Further, it is the fastest domain adaptation algorithm in terms of computational complexity, while maintaining its excellent performance by using only a

subset of source and target data. It determines a target subspace representation for both domains and transfers target basis information to source data via Nyström Singular Value Decomposition (SVD) with class-wise sampling. Further, we show that Nyström approximation is well suited for non-square landmark matrices, in contrast to common decomposition (Nemtsov et al., 2016) by applying post-processing.

The rest of the paper is organized as follows: We give an overview of related work in section 2. The underlying mathematical concepts are given in section 3. The proposed approach is discussed in section 4, followed by an experimental part in section 5, addressing the classification performance and computational time. A summary with a discussion of open issues is provided in the conclusion at the end of the paper. More detailed derivations, description of experiments, implementation details and additional results can be found in the appendix.

## 2    RELATED WORK

In general, homogeneous transfer learning (Weiss et al., 2016) approaches, or domain adaptation (DA), distinguish roughly between the following strategies: Methods implementing the *symmetric feature adaptation* (Weiss et al., 2016) are trying to find a common latent subspace for source and target domain to reduce marginal distribution differences, such that the underlying structure of the data is preserved in the subspace. A baseline approach for symmetric feature adaptation is the Transfer Component Analysis (TCA) (Pan et al., 2011). TCA finds a suitable subspace transformation called transfer components via minimizing the Maximum Mean Discrepancy (MMD) in the Reproducing Kernel Hilbert Space (RKHS). The Geodesic Flow Kernel (GFK) finds a particular subspace by embedding original data onto a Grassmannian manifold, matches source and target on the geodesic flow and determines a suitable domain adaptation kernel by applying the kernel trick (Gong et al., 2012).
The *relational-knowledge adaptation* aims to find some relationship between source and target data (Weiss et al., 2016). Transfer Kernel Learning (TKL) (Long et al., 2015) is a recent approach, which approximates a kernel of training data $K(\mathbf{Z})$ by test kernel $K(\mathbf{X})$ via the Nyström kernel approximation.

However, recent studies conclude that the asymmetric transformation of source data is a promising approach in domain adaptation due to the better classification performance (Sun et al., 2016; Fernando et al., 2016; Raab & Schleif, 2018; Long et al., 2013). Possibly, because a classifier is trained on the source label information in the target domain (Sun et al., 2016). The good performance has led to a variety of unsupervised asymmetric domain adaptation algorithms (Elhadji-Ille-Gado et al., 2017; Shao et al., 2014; Blitzer et al., 2011; Thopalli et al., 2019), to name a few. The *asymmetric feature adaptation* approaches try to transform source domain data in the target (subspace) domain to match the source to target subspace and is most related to our work. In comparison to the symmetric feature adaptation approaches, there is no shared subspace, but only the target space (Weiss et al., 2016). The Joint Distribution Adaptation (JDA) (Long et al., 2013) solves divergences in marginal distributions similar to TCA, but aligning conditional distributions with pseudo-labeling techniques. The Subspace Alignment (SA) (Fernando et al., 2013) computes a target subspace representation based on the correlation between target and source subspace projectors and hence aligning source and target data. The Correlation Alignment (CORAL) (Sun et al., 2016) technique transfers second-order statistics of the target domain into whitened source data.
Given these methods, there is an overall trend in domain adaptation in the direction of non-linear problem formulations, indirect alignment of matrix differences and learning a suitable subspace for the source domain in the target domain and have similar problem statements to ours. *However*, in this work, we do not learn a suitable source subspace transformation, but explicitly override the structural information of source domain with the target one. With this, we model the source subspace domain as part of the target subspace.
The considered domain adaptation methods have approximately a complexity of $\mathcal{O}(n^2)$ where $n$ is the most significant number of samples concerning target or source. These algorithms pursue *transductive* adaptation (Pan & Yang, 2010), because some *unlabeled test* data *must* be available at training time. These transfer-solutions cannot be directly used as predictors, but instead, are wrappers for classification algorithms. The respectively used baseline classifier is the Support Vector Machine (SVM).

## 3 PRELIMINARIES

We introduce the basics of the Nyström kernel approximation in section 3.1, which is the foundation of the Nyström based Singular Value Decomposition in section 3.2. The Nyström-SVD is used for creating the subspace transformation of Nyström Basis Transfer in section 4.1.

### 3.1 NYSTRÖM APPROXIMATION

The computational complexity of calculating kernels or eigensystems scales with $\mathcal{O}(n^3)$ where $n$ is the sample size (Williams & Seeger, 2001). Therefore, low-rank approximations and dimensionality reduction of data matrices are popular methods to get a better computational performance. In this scope, however not limited to it, the Nyström approximation (Williams & Seeger, 2001) is a reliable technique to accelerate eigendecomposition or approximation of general symmetric matrices (Gisbrecht & Schleif, 2015).

It computes an approximated set of eigenvectors and values based on a usually much smaller sample matrix. The landmarks are typically picked at random, but advanced sampling concepts could be used as well (Talwalkar et al., 2012). The approximation is exact if the sample size is equal to the rank of the original matrix and the rows of the sample matrix are linear independent (Gisbrecht & Schleif, 2015).

In general, the Nyström approximation technique assumes a symmetric matrix $\mathbf{K} \in \mathbb{R}^{n \times n}$ with a decomposition of the form

$$\mathbf{K} = \begin{bmatrix} \mathbf{A} & \mathbf{B} \\ \mathbf{C} & \mathbf{D} \end{bmatrix}, \tag{1}$$

with $\mathbf{A} \in \mathbb{R}^{s \times s}$, $\mathbf{B} \in \mathbb{R}^{s \times (n-s)}$, $\mathbf{C} \in \mathbb{R}^{(n-s) \times s}$ and $\mathbf{D} \in \mathbb{R}^{(n-s) \times (n-s)}$. The submatrix $\mathbf{A}$ is called the landmark matrix containing $s$ randomly chosen rows and columns from $\mathbf{K}$ and has the Eigenvalue Decomposition (EVD) $\mathbf{A} = \mathbf{U}\boldsymbol{\Lambda}\mathbf{U}^{-1}$. Where eigenvectors are $\mathbf{U} \in \mathbb{R}^{s \times s}$ and eigenvalues are on the diagonal of $\boldsymbol{\Lambda} \in \mathbb{R}^{s \times s}$. The remaining approximated eigenvectors $\hat{\mathbf{U}}$ of $\mathbf{K}$ as the part $\mathbf{C}$ or $\mathbf{B}^T$, are obtained by the Nyström method with $\hat{\mathbf{U}}\boldsymbol{\Lambda} = \mathbf{C}\mathbf{U}$. Combining $\mathbf{U}$ and $\hat{\mathbf{U}}$ the *full* approximated eigenvectors of $\mathbf{K}$ are

$$\tilde{\mathbf{U}} = \begin{bmatrix} \mathbf{U} \\ \hat{\mathbf{U}} \end{bmatrix} = \begin{bmatrix} \mathbf{U} \\ \mathbf{C}\mathbf{U}\boldsymbol{\Lambda}^{-1} \end{bmatrix} \in \mathbb{R}^{n \times s}. \tag{2}$$

The right part of the EVD ($\tilde{\mathbf{U}}^{-1}$) of $\mathbf{K}$ can be obtained via Nyström similar to equation 2 by

$$\tilde{\mathbf{V}} = \begin{bmatrix} \mathbf{U}^{-1} & \boldsymbol{\Lambda}^{-1}\mathbf{U}^{-1}\mathbf{B} \end{bmatrix}. \tag{3}$$

Combining equation 2, equation 3 and $\boldsymbol{\Lambda}$, the matrix $\mathbf{K}$ is approximated by

$$\tilde{\mathbf{K}} = \tilde{\mathbf{U}}\boldsymbol{\Lambda}\tilde{\mathbf{V}} = \begin{bmatrix} \mathbf{U} \\ \mathbf{C}\mathbf{U}\boldsymbol{\Lambda}^{-1} \end{bmatrix} \boldsymbol{\Lambda} \begin{bmatrix} \mathbf{U}^{-1} & \boldsymbol{\Lambda}^{-1}\mathbf{U}^{-1}\mathbf{B} \end{bmatrix}. \tag{4}$$

The Nyström approximation error is given by the Frobenius Norm between ground truth and reconstructed matrices, i.e. $E_{ny} = ||\tilde{\mathbf{K}} - \mathbf{K}||_F$, with bounds proven by Gittens & Mahoney (2013).

### 3.2 GENERAL MATRIX APPROXIMATION

Another application of the Nyström method is the approximation of the Singular Value Decomposition, which generalizes the concept of matrix decomposition with the consequence that respective matrices must not be squared.

Let $\mathbf{K} \in \mathbb{R}^{n \times d}$ be a rectangular matrix with the decomposition as in equation 1. The SVD of the landmark matrix is given by $\mathbf{A} = \mathbf{L}\mathbf{S}\mathbf{R}^T$ where $\mathbf{L}$ are left and $\mathbf{R}$ are right singular vectors. $\mathbf{S}$ are positive singular values. The left and right singular vectors for the non-symmetric part $\mathbf{C}$ and $\mathbf{B}$ are obtained via Nyström techniques and are defined as $\hat{\mathbf{L}} = \mathbf{C}\mathbf{R}\mathbf{S}^{-1}$ and $\hat{\mathbf{R}} = \mathbf{B}^T\mathbf{L}\mathbf{S}^{-1}$ respectively (Nemtsov et al., 2016).

Applying the same principal as for Nyström-EVD, $\mathbf{K}$ is approximated by

$$\tilde{\mathbf{K}} = \tilde{\mathbf{L}}\mathbf{S}\tilde{\mathbf{R}}^T = \begin{bmatrix} \mathbf{L} \\ \hat{\mathbf{L}} \end{bmatrix} \mathbf{S} \begin{bmatrix} \mathbf{R} & \hat{\mathbf{R}} \end{bmatrix} = \begin{bmatrix} \mathbf{L} \\ \mathbf{C}\mathbf{R}\mathbf{S}^{-1} \end{bmatrix} \mathbf{S} \begin{bmatrix} \mathbf{R} & \mathbf{S}^{-1}\mathbf{L}^T\mathbf{B} \end{bmatrix}. \tag{5}$$

### 3.3 POLAR DECOMPOSITION

The Polar decomposition (Higham, 2005) theorem is used to validate the proposed subspace transformations as PCA transformations in section 4 and the class-wise sampling strategy in section 4.2. It is a universal decomposition applicable to an arbitrary matrix and is defined as $\mathbf{X} = \mathbf{QP}$. Where $\mathbf{Q} = \mathbf{LR}^T$ and $\mathbf{P} = \mathbf{RSR}^T$ with $\mathbf{S}$ as singular values, $\mathbf{L}$ and $\mathbf{R}$ are left and right singular vectors respectively. If $\mathbf{X}$ is a square matrix, the decomposition is unique and $\mathbf{U}$ is orthogonal and a rotation matrix. $\mathbf{P}$ is positive semi-definite and contains the scaling factors of $\mathbf{X}$.

**Theorem 1** (Reusing of Eigensystem). *Let $\mathbf{X} \in \mathbb{R}^{n \times d}$ and $\mathbf{K} = \mathbf{XX}^T$ with EVD of $\mathbf{K} = \mathbf{U \Lambda U}^{-1}$ and Singular Value Decomposition (SVD) of $\mathbf{K} = \mathbf{LSR}^T$. Taking the square root of $\mathbf{K}$ and using the Polar Decomposition with*

$$\mathbf{K}^{\frac{1}{2}} = (\mathbf{U \Lambda U}^{-1})^{\frac{1}{2}} = (\mathbf{QP})^{\frac{1}{2}} = (\mathbf{LR}^T \mathbf{RSR}^T)^{\frac{1}{2}} = (\mathbf{LSR}^T)^{\frac{1}{2}} = \mathbf{LS}^{\frac{1}{2}} \mathbf{R}^T, \tag{6}$$

*then the eigenvectors and square root eigenvalues of $\mathbf{K}$ are singular vectors and values of $\mathbf{X}$ respectively.*

### 3.4 GERSCHGORIN THEOREM

The Gerschgorin theorem (Varga, 2004) provides a geometric structure to bound eigenvalues to so-called discs for complex square matrices, but also generalize to none complex square matrices. The work of Qi (1984) expands the Gerschgorin circles to so-called Gerschgorin type circles for singular values:

**Theorem 2** (Gerschgorin Type Bound for Singular Values (Qi, 1984)). *Given the matrix $\mathbf{X} \in \mathbb{R}^{n \times d}$ with $n, d \geq 1$, the singular values $\{s_i\}_{i=1}^n$ of $\mathbf{X}$ are in the range of*

$$s_i = \{p_i \pm |r_i|\}, \quad i = 1, \dots, n. \tag{7}$$

*Where $p_i = |x_{ii}|$ and the range $r_i$ is defined as*

$$r_i = \max \left\{ \sum_{j=1, j \neq i}^d |x_{ij}|, \ \sum_{j=1, j \neq i}^n |x_{ji}| \right\}, \quad i = 1, \dots, n. \tag{8}$$

By this, it is possible to estimate the numerical range of singular values of $\mathbf{X}$ and is used in the error bound given in section 4.3.

## 4 BASIS TRANSFER

The task of domain adaptation is to align distribution differences. That means that underlying statistics will be similar afterwards. The key idea of the approach presented in the following is that explicit alignment of data matrices without any knowledge of statistics and distribution of data is possible and successful. It is based on the work of Schölkopf et al. (1998), which proposed that if two kernel matrices are similar, they follow similar distributions. Hence, we make the domain data as similar as possible by overriding the syntactic structure, *which will result in similar kernels*. This also applies to arbitrary rectangular data matrices of a linear kernel, because it can be reconstructed by using theorem 1. Hence, the method is also applicable in the original feature space without requiring a positive semi definite kernel.

The recent Basis-Transfer (BT) (Raab & Schleif, 2018) approach already showed great transfer capabilities and performance by aligning $\mathbf{X} \in \mathbb{R}^{n \times d}$ and $\mathbf{Z} \in \mathbb{R}^{m \times d}$ with a small error in terms of the Frobenius norm. *However, this required equal samples sizes $n = m$.* In BT, the following optimization problem was considered:

$$\min_{\mathbf{M}, \mathbf{T}} \|\mathbf{MZT} - \mathbf{X}\|_F. \tag{9}$$

Where $\mathbf{M}$ and $\mathbf{T}$ are transformation matrices, drawing the source ($\mathbf{Z}$) closer to target ($\mathbf{X}$) data. A solution of equation 9 is found in closed-form, summarized in three steps (Raab & Schleif, 2018): First, normalize data to standard mean and variance. It is assumed that this will align marginal distributions in Euclidean space without considering label information (Raab & Schleif, 2018). Second, compute an SVD of source and target data, i.e. $\mathbf{Z} = \mathbf{L}_Z \mathbf{S}_Z \mathbf{R}_Z^T$ with $\mathbf{L}_Z \in \mathbb{R}^{m \times m}$ and

$\mathbf{R}_Z \in \mathbb{R}^{d \times d}$. And $\mathbf{X} = \mathbf{L}_X \mathbf{S}_X \mathbf{R}_X^T$ with $\mathbf{L}_X \in \mathbb{R}^{n \times n}$ and $\mathbf{R}_X \in \mathbb{R}^{d \times d}$. Note that we denote $\odot_Z / \odot_X$ as source/target related matrix respectively. Next, the approach assumes $\mathbf{S}_Z \sim \mathbf{S}_X$ in terms of the Frobenius norm due to similar domains providing similar scaling factors by singular values and normalization. Finally, compute a solution for equation 9 by solving the linear equations. One obtains $\mathbf{M} = \mathbf{L}_X \mathbf{L}_Z^{-1}$ and $\mathbf{T} = \mathbf{R}_Z^{-1} \mathbf{R}_X^T$. Note that $\mathbf{L}_Z \mathbf{L}_Z^{-1} = \mathbf{R}_Z \mathbf{R}_Z^{-1} = \mathbf{I}$. Finally, a transfer operation is applied such that the source matrix is approximated by using the available target information by

$$\tilde{\mathbf{Z}} = \mathbf{MZT} = \mathbf{L}_X \mathbf{L}_Z^{-1} \mathbf{L}_Z \mathbf{S}_Z \mathbf{R}_Z \mathbf{R}_Z^{-1} \mathbf{R}_X^T = \mathbf{L}_X \mathbf{S}_Z \mathbf{R}_X^T. \tag{10}$$

With $\tilde{\mathbf{Z}} \in \mathbb{R}^{m \times d}$ as approximated source data, used for training.

In the following, the work (Raab & Schleif, 2018) is hereby substantially improved and we propose a Nyström based version with following improvements: Reduction of computational complexity, neglecting sample size requirements and achieve a low-rank projection via Nyström approximation with class-wise sampling and implicit dimensionality reduction.

Recap equation 9 and consider the slightly changed optimization problem

$$\min_{\mathbf{M}} ||\mathbf{MZ} - \mathbf{X}||_F. \tag{11}$$

Where a transformation matrix $\mathbf{M}$ must be found, which is again obtained in closed-form. Note this is a transfer learning interpretation of the orthogonal procrustes problem (Schönemann, 1966). Because we apply a dimensionality reduction technique, just the left-sided transformation matrix must be determined, derived as follows: Based on the relationship between SVD and EVD, the Principal Component Analysis (PCA) can be rewritten in terms of SVD. Consider the target matrix with SVD:

$$\mathbf{X}^T \mathbf{X} = (\mathbf{R}_X \mathbf{S}_X \mathbf{L}_X^T)(\mathbf{L}_X \mathbf{S}_X \mathbf{R}_X^T) = \mathbf{R}_X \mathbf{S}_X^2 \mathbf{R}_X^T. \tag{12}$$

Where $\mathbf{R}_X \in \mathbb{R}^{d \times s}$ as eigenvectors and $\mathbf{S}_X^2 \in \mathbb{R}^{s \times s}$ as eigenvalues of $\mathbf{X}^T \mathbf{X}$. By choosing only the biggest $s$ eigenvalues and corresponding eigenvectors the dimensionality of $\mathbf{X}$ is reduced by

$$\mathbf{X}_s = \mathbf{X} \mathbf{R}_s = \mathbf{L}_s \mathbf{S}_s \mathbf{R}_s^T \mathbf{R}_s = \mathbf{L}_s \mathbf{S}_s. \tag{13}$$

With $\mathbf{L}_s \in \mathbb{R}^{n \times s}$, $\mathbf{S}_s \in \mathbb{R}^{s \times s}$. $\mathbf{X}_s \in \mathbb{R}^{n \times s}$ is the reduced target matrix. Hence, only a left sided transformation in equation 11 is required, because the right sided transformation is omitted in equation 13. Note, that for equation 12 a linear covariance or kernel is used.

However, this procedure requires a complete data matrix or corresponding singular values and scales the in worst case with $\mathcal{O}(n^3)$ (Williams & Seeger, 2001). In the following, we apply Nyström-SVD and show that only a subset of the data is required, which simultaneously reduces computational complexity and eliminates the need to examine all singular values.

## 4.1 Nyström Basis Transfer

Let $\mathbf{Z}$ and $\mathbf{X}$ have a valid decomposition given as in equation 1. Note for clarity the Nyström notation is used as in section 3.2. For a Nyström-SVD we sample from both matrices $s$ values obtaining landmarks matrices $\mathbf{A}_Z = \mathbf{L}_Z \mathbf{S}_Z \mathbf{R}_Z^T \in \mathbb{R}^{s \times s}$ and $\mathbf{A}_X = \mathbf{L}_X \mathbf{S}_X \mathbf{R}_X^T \in \mathbb{R}^{s \times s}$. Based on Nyström-SVD in equation 5, the dimensions are reduced as in equation 13 keeping only most relevant data structures

$$\mathbf{X}_s = \tilde{\mathbf{L}}_X \mathbf{S}_X = \begin{bmatrix} \mathbf{L}_X \\ \hat{\mathbf{L}}_X \end{bmatrix} \mathbf{S}_X = \begin{bmatrix} \mathbf{L}_X \\ \mathbf{C}_X \mathbf{R}_X \mathbf{S}_X^{-1} \end{bmatrix} \mathbf{S}_X \in \mathbb{R}^{n \times s}. \tag{14}$$

Hence, it is sufficient to only compute a SVD of $\mathbf{A}_X$ instead of $\mathbf{X}$ with $s \ll m, d$ and therefore is considerably lower in computational complexity. Analogy, we approximate source data by $\mathbf{Z}_s = \tilde{\mathbf{L}}_Z \mathbf{S}_Z \in \mathbb{R}^{n \times s}$. Since we again assume $\mathbf{S}_Z \sim \mathbf{S}_x$ due to data normalization and domain similarities, solving the linear equation as a possible solution for equation 11, leads to $\mathbf{M} = \tilde{\mathbf{L}}_X \tilde{\mathbf{L}}_Z^{-1}$. Plugging it back we obtain

$$\mathbf{Z}_s = \tilde{\mathbf{L}}_X \tilde{\mathbf{L}}_Z^{-1} \tilde{\mathbf{L}}_Z \mathbf{S}_Z = \tilde{\mathbf{L}}_X \mathbf{S}_Z \in \mathbb{R}^{n \times s}. \tag{15}$$

Where again a basis of a target subspace transfers structural information into the source domain. The matrix $\mathbf{Z}_s$ is used for training and $\mathbf{X}_s$ is used for testing. We showed in equation 13, that equation 14 and equation 15 are valid PCA transformations. By definition of SVD follows $\tilde{\mathbf{L}}_Z \tilde{\mathbf{L}}_Z^T = \tilde{\mathbf{L}}_X \tilde{\mathbf{L}}_X^{-1} = \mathbf{I}$ and $\tilde{\mathbf{L}}_X$ is an orthogonal basis. Therefore, equation 14 and equation 15 are orthogonal transformations. In particular, equation 15 transforms the source data into the target subspace and

projects it onto the principal components of $\mathbf{X}$. If data matrices $\mathbf{X}$ and $\mathbf{Z}$ are standardized[1], the geometric interpretation is a rotation of source data w.r.t to angles of the target basis. The relationship between BT and NBT is in the number of samples $s$ used by Nyström. If the rank of the data matrices is equal to $s$ the approximation is exact and it became a subspace version of BT.

According to (Weiss et al., 2016), it is an asymmetric transfer approach. Further, it is transductive (Pan & Yang, 2010), *where unlabeled target data is needed at training time*. Subsequently this approach is denoted as Nyström Basis Transfer (NBT).

But uniform sampling may not be optimal for Nyström, given labeled data in a classification task (Schleif et al., 2018). Therefore, we integrate class-wise sampling in the following.

## 4.2 SAMPLING STRATEGY FOR NYSTRÖM

The standard technique to create Nyström landmark matrices is to sample uniform or find clusters in the data matrix (Talwalkar et al., 2012). In supervised learning, sampling should utilize class-wise sampling to properly include class-depending attributes of a matrix into the approximation (Schleif et al., 2018). This is especially necessary for $||\mathcal{Y}|| > 2$. However, a decomposition as in equation 1, required for Nyström-SVD is intractable with class-wise sampling, because respective matrices are non-square: Let $\mathbf{Z} \in \mathbb{R}^{m \times d}$ with $m \neq d$ and landmark indices $I = \{i_1, \ldots, i_s\}$ with at least one $i_j > d$, then it is by definition undefined and is especially true for $n > d$. Therefore, we sample rows class-wise forming $\mathbf{A}_Z^d \in \mathbb{R}^{s \times d}$ instead of $\mathbf{A}_Z \in \mathbb{R}^{s \times s}$. Using the insights of theorem 1, we implicitly obtain an SVD of a *square* landmark matrix, i.e. $\mathbf{A}_Z^d \mathbf{A}_Z^{dT} = \mathbf{K}_Z^d \in \mathbb{R}^{s \times s}$ by computing $\mathbf{A}_Z^d = \mathbf{L}_Z \mathbf{S}_Z^d \mathbf{R}_Z^T$. Hence, we can proceed without a non-square landmark matrix and using the SVD of $\mathbf{A}_Z^d$ analogy as in equation 15. Therefore, it is possible to sample from the whole range of source data and by application of the Polar decomposition, the standard decomposition as in equation 1 is not required. The resulting singular values and vectors are utilized for successive Nyström approximations.

The sampling from test data $\mathbf{X}$ is done uniformly row-wise, because of missing class information and SVD is obtained analogy via $\mathbf{A}_X^d = \mathbf{L}_X \mathbf{S}_X^d \mathbf{R}_X^T$ and used in equation 14.

However, the possible range of singular values of $\mathbf{A}_Z^d$ is naturally not equal to $\mathbf{A}_Z$. Utilizing theorem 2 it is easy to show that $\mathbf{S}_Z^d \neq \mathbf{S}_Z$. It scales approximated matrices $\mathbf{Z}_s$ (equation 15) different through $\mathbf{S}_Z^d$ and accurate normalization of matrix cannot be guaranteed. Therefore, we apply a post-processing correction and apply z-normalization against the standard convention *after* NBT. The singular vectors also have an approximation error. However, both subspace projections are based on the same transformation matrix, hence making an identical error and as a result, the error should not affect the classification. The process is given in the figure 2 (appendix A.2) and showing the alignment of source to the target domain. Further, it shows the mechanics of the approach: The structural similarities stay the same, while the scaling is changed leading to a high similarity after the post-processing correction.

## 4.3 PROPERTIES OF NYSTRÖM BASIS TRANSFER

The computational complexity of Nyström Basis Transfer (NBT) is composed of economy-size SVD of landmark matrices $\mathbf{A}_Z^d$ and $\mathbf{A}_X^d$ with complexity $\mathcal{O}(2s^2)$. The matrix inversion of diagonal matrix $\mathbf{S}_X^{d\,-1}$ in equation 14 can be neglected. Remaining $k$ matrix multiplications are $\mathcal{O}(ks^2)$ contributing to the overall complexity of NBT which is $O(s^2)$ with $s \ll n, m, d$. This makes NBT the fastest domain adaptation solution in terms of computational complexity in comparison to discussed methods in section 2.

The difference between source and target domain after NBT, i.e. approximation error of source by target domain is bounded by

$$E_{NBT} = \|\mathbf{X}_s - \mathbf{Z}_s\|_F \leq |\sqrt{n}| |D_X^s - D_Z^s| < \|\mathbf{X} - \mathbf{Z}\|_F. \tag{16}$$

Where $||\tilde{\mathbf{L}}_X|| \leq \sqrt{n}$ is the bounding of the normalized singular vectors and $D_{(\cdot)}$ is the maximum sum of the $s$ singular values obtained by the numerical estimation, given in theorem 2, of $\mathbf{X}_s$ and $\mathbf{Z}_s$ respectively. The equation 16 shows that NBT has a lower norm compared to BT or original data and proofs that the matrices are aligned during NBT, reducing the distribution differences. The

---

[1]Experimental data are standardized to mean zero and variance one in the preprocessing.

respective proof is given in appendix A.1.

The out-of-sample extension for unseen target/source samples, e. g. $\mathbf{x} \in \mathbb{R}^d$, is analog to equation 14. Based on equation 13 a subspace projection via (approximated) right singular vectors is also valid. Hence, a sample can be projected into the subspace via $\mathbf{x}_s = \mathbf{x}\tilde{\mathbf{R}}_X^T = \mathbf{x} \left[ \mathbf{R}_X \quad \mathbf{S}_X^{d\ -1}\mathbf{L}^T\mathbf{B}_X \right]$ and be evaluated by an arbitrary classifier learned in the subspace. The pseudo code of NBT is shown in Algorithm 1.

---

**Algorithm 1** Nyström Basis Transfer (NBT)

---

**Require:** $\mathbf{Z}$ as $m$ sized training; $\mathbf{X}$ as $n$ sized test set; $\mathbf{Y}$ as $m$ sized training label vector; $s$ as number of landmarks parameter.
**Ensure:** New Source $\mathbf{Z_s}$; new Target $\mathbf{X}_s$;
1: $\mathbf{A}_X^d, \mathbf{A}_Z^d, \mathbf{C}_X =$ matrix_decomposition($\mathbf{X},\mathbf{Z},s$)            $\triangleright$ According to section 4.2
2: $\mathbf{S}_Z^d = SVD(\mathbf{A}_Z^d)$;                                                                  $\triangleright$ SVD of landmark matrix of $\mathbf{Z}$
3: $\mathbf{L}_X, \mathbf{S}_X^d, \mathbf{R}_X = SVD(\mathbf{A}_X^d)$;                                        $\triangleright$ SVD of landmark matrix of $\mathbf{X}$
4: $\tilde{\mathbf{L}}_X = \left[ \mathbf{L}_X \quad \mathbf{C}_X\mathbf{R}_X\mathbf{S}_X^{d\ -1} \right]^T$     $\triangleright$ According to equation 14
5: $\mathbf{X}_s = \tilde{\mathbf{L}}_X\mathbf{S}_X^d$                                                         $\triangleright$ According to equation 14
6: $\mathbf{Z}_s = \tilde{\mathbf{L}}_X\mathbf{S}_Z^d$                                                         $\triangleright$ According to equation 15
7: $\mathbf{Z}_s, \mathbf{X}_s =$ z_normalization($\mathbf{Z}_s,\mathbf{X}_s$)                                $\triangleright$ Per Matrix; Effect shown in figure 2

---

## 5 EXPERIMENTS

We follow the experimental design typical for domain adaptation algorithms (Long et al., 2015; Gong et al., 2012; Long et al., 2013; Pan & Yang, 2010; Pan et al., 2011; Sun et al., 2016; Mahadevan et al., 2019; Fernando et al., 2013). A crucial characteristic of datasets for domain adaptation is that domains for training and testing are different but related. This relation exists because the train and test classes have the same top category or source. The classes itself are subcategories or subsets. The parameters for respective methods[2] are determined for best performance in terms of accuracy via grid search. A detailed description of experiments can be found in appendix A.4. In the experiments, the SVM independent of being a baseline or underlying classifier for domain adaptation methods uses the RBF-Kernel. The tests are conducted on the datasets groups Reuters, Newsgroup and Office-Caltech. A detailed dataset description is given in appendix A.3.

### 5.1 PERFORMANCE COMPARISON

The experiments are cumulated as three dataset group, which are Reuters, Newsgroup and Office-Caltech. The results are summarized in table 1. It shows the mean errors of the experiments per dataset group. To determine statistically significant differences, the two-sided Kolmogorov-Smirnoff-Test with a p-value of $1\%$ is applied. The $*$ marks statistical significance compared to NBT. The NBT is compared to baseline SVM and standard domain adaptation methods.

The NBT and BT methods have excellent performance and outperform every other algorithm by far and are overall similar. Looking at the datasets individually, the only competitive approach is SA, however, in the overall rating, all non-BT methods are statistically worse.

Note that BT and NBT are very similar at Office-Caltech, this is caused because within NBT the rank $r$ of a given image data matrices is equal to the number of samples drawn by the Nyström approximation. Hence, the approximation is exact and we use the most important $r$ singular vectors and values. BT and NBT are overall much better than remaining approaches, caused by following possible reasons: Aligning differences of Frobenius norm between domains using a target (subspace)-transformation is sufficient as a problem statement and overriding of source basis is a successful approach. Further, by Schölkopf et al. (1998) it will also align differences in subsequent kernels. Explicit integration of non-linearity or kernel statistic alignments does not necessarily lead to more knowledge transfer. But, second-order statistics *but also* structural and geometric information should be transferred.

---

[2] Source code and datasets are hosted at https://github.com/iclr-nbt/nbt.git

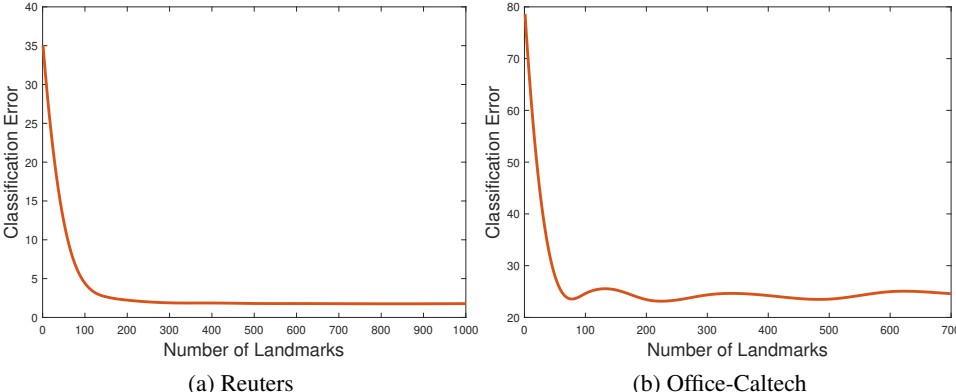

(a) Reuters                    (b) Office-Caltech

Figure 1: Relationship between number of landmarks and mean classification error over all Reuters datasets, shown in left figure and mean over all Office-Caltech datasets, shown in right figure.

| Dataset | SVM | TCA 2011 | JDA 2013 | TKL 2015 | GFK 2012 | SA 2013 | CORAL 2016 | BT 2018 | NBT |
|---|---|---|---|---|---|---|---|---|---|
| Reuters | 32.1∗ | 31.8∗ | 33.1∗ | 23.2∗ | 35.0∗ | 7.6 | 35.7∗ | 3.5 | **1.9** |
| Newsgroup | 17.8∗ | 14.7 ∗ | 17.4∗ | 6.7 | 22.5∗ | **2.3** | 28.1 ∗ | 2.6 | **2.3** |
| Office-Caltech | 55.5 ∗ | 51.5 ∗ | 51.4 ∗ | 49.2 ∗ | 55.3 ∗ | 68.7 ∗ | 52.6 ∗ | **20.7** | 20.8 |
| Overall Mean | 35.1 ∗ | 32.7 ∗ | 34.0 ∗ | 26.4 ∗ | 37.6 ∗ | 26.2 ∗ | 38.8 ∗ | 8.90 | **8.3** |

Table 1: Result of experiments as mean error in percent over each dataset group. Bold marks winner. ∗ marks statistical differences with a p-value of $1\%$ against NBT. The standard deviation was omitted, since it is the same for all and on average around 2-3%.

The sensitivity of the number of landmarks on prediction error as the only parameter of NBT is demonstrated in figure 1. It shows a comparison of the number of landmarks and the mean classification error over Reuters and Office-Caltech datasets. At Reuters, it is a monotonically decreasing convex function with decreasing error by increasing the number of landmarks. This supports the Nyström error expectation of approximating the real rank of a matrix. However, the image function has many local minima. We assume this indicates that only a subset of features is relevant and correlate for classification and remaining features are noise. A time comparison is given in the appendix A.5 showing that our NBT approach is the overall fastest domain adaptation approach.

## 6  CONCLUSION

We proposed a low-rank domain approximation algorithm called Nyström Basis Transfer. It transfers source and target data into the target subspace, requiring only a subset of domain data from both domains. The dimensionality reduction, paired with smart class-wise sampling showed its reliability and robustness in this study. Validated on common domain adaptation tasks and data, it showed an outstanding performance both in absolute and statistical values. Additionally, it is lowest in computational complexity compared to discussed solutions. NBT is an extension of earlier versions of Basis Transfer via Nyström Methods and is a low-rank and fast domain adaption solution.
In future work, deep transfer learning, real-world and other domain adaptation datasets should be considered. A more comprehensive discussion of the Nyström approximation error with the proposed decomposition should be theoretically evaluated and compared to current Nyström techniques.

---

[3]We encountered numerical issues at our experiments, therefore the shown results are taken from (Mahadevan et al., 2019) with a similar sampling scheme.

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

## A    APPENDIX

### A.1    NORM BOUND OF NYSTRÖM BASIS TRANSFER

This section discusses the approximation error of Nyström Basis Transfer in terms of the Frobenius norm of the matrix difference shown in equation 16. First, we clarify some definitions used in the subsequent theorem. In this work, we define the Frobenius norm of a matrix $\mathbf{X} \in \mathbb{R}^{n \times d}$ as $\|\mathbf{X}\|_F = \sqrt{\sum_{i=1}^{n} \sum_{j=1}^{d} |\mathbf{x}_{ij}|^2}$. By using theorem 2 we can bound the norm of the singular values $\|\mathbf{S}\|_F$ of $\mathbf{X}$ by the square root of the squared sum of the numerical range given by

$$D = \|\mathbf{S}\|_F \leq \sqrt{\sum_{i=1}^{n} (p_i + |r_i|)^2}. \tag{17}$$

Finally, we assume that the non-zero singular vectors are normalized and have by definition unit length. Therefore, the norm of a singular vector matrix with $n$ singular vectors can be bounded by $\sqrt{n}$.

**Theorem 3.** *Given two rectangular matrices* $\mathbf{X}, \mathbf{Z} \in \mathbb{R}^{n \times d}$ *with* $n, d > 1$ *and* $rank(\mathbf{X}), rank(\mathbf{Z}) > 1$. *The norm* $\|\mathbf{Z}_s - \mathbf{X}_s\|_F$ *in the subspace* $\mathbb{R}^s$ *induced by normalized singular vectors* $\mathbf{L} \in \mathbb{R}^{n \times s}$ *is bounded by*

$$E_{NBT} = \|\mathbf{Z}_s - \mathbf{X}_s\|_F \leq \sqrt{n}\|D_X^s - D_Z^s\| < \|\mathbf{Z} - \mathbf{X}\|_F. \tag{18}$$

*Proof.* We start by proofing the first part given by $\|\mathbf{Z}_s - \mathbf{X}_s\|_F \leq \sqrt{n}\|D_X^s - D_Z^s\|$. Using equation 14, equation 15 and equation 16 the respective term can be rewritten in terms of a SVD-PCA using only the biggest $s$ singular values. Further, substituting $\mathbf{M} = \mathbf{L}_X \mathbf{L}_Z^{-1}$ leads to

$$\|\mathbf{Z}_s - \mathbf{X}_s\|_F = \|\mathbf{M}\mathbf{Z} - \mathbf{X}_s\|_F = \left\|\mathbf{L}_X \mathbf{L}_Z^{-1} \mathbf{L}_Z \mathbf{S}_Z - \mathbf{L}_X \mathbf{S}_X\right\|_F = \|\mathbf{L}_X \mathbf{S}_Z - \mathbf{L}_X \mathbf{S}_X\|_F. \tag{19}$$

Where we assume $1 \leq s < n$. This can be simplified applying the Cauchy-Schwarz inequality twice with

$$\|\mathbf{L}_X \mathbf{S}_Z - \mathbf{L}_X \mathbf{S}_X\|_F \leq \|\mathbf{L}_X\|_F \|\mathbf{S}_Z - \mathbf{S}_X\|_F \leq \|\mathbf{L}_X\|_F (\|\mathbf{S}_Z\|_F - \|\mathbf{S}_X\|_F). \tag{20}$$

The norm of the singular values $\mathbf{S}_Z, \mathbf{S}_X \in \mathbb{R}^{s \times s}$ can be bounded by equation 17 resulting in

$$\|\mathbf{S}_Z\|_F \leq D_Z^s = \sqrt{\sum_{i=1}^{s}(p_i^z + |r_i^z|)^2}, \quad \|\mathbf{S}_X\|_F \leq D_X^s = \sqrt{\sum_{i=1}^{s}(p_i^x + |r_i^x|)^2}. \tag{21}$$

Where $p_i^z$ and $r_i^z$ are the bound of a singular value of $\mathbf{Z}_s$ and $p_i^x$ and $r_i^x$ are the bound of a singular value of $\mathbf{X}_s$.
Further, the singular vectors $\mathbf{L}_X$ are normalized and thus can be bounded by $\|\mathbf{L}_X\|_F \leq \sqrt{n}$. Applying both to equation 20 leads to

$$\|\mathbf{L}_X \mathbf{S}_Z - \mathbf{L}_X \mathbf{S}_X\|_F \leq \|\mathbf{L}_X\|_F (\|\mathbf{S}_Z\|_F - \|\mathbf{S}_X\|_F) \leq \sqrt{n}\|D_X^s - D_Z^s\|. \tag{22}$$

The second term $\sqrt{n}\|D_X^s - D_Z^s\| < \|\mathbf{Z} - \mathbf{X}\|_F$ is obtained analogy. Given the SVD of the original matrices we obtain

$$\|\mathbf{Z} - \mathbf{X}\|_F = \|\mathbf{L}_Z \mathbf{S}_Z \mathbf{R}_Z - \mathbf{L}_X \mathbf{S}_X \mathbf{R}_X\|_F. \tag{23}$$

By again using the Cauchy-Schwarz inequality we obtain

$$\|\mathbf{Z} - \mathbf{X}\|_F \leq \|\mathbf{L}_Z\|_F \|\mathbf{S}_Z\|_F \|\mathbf{R}_Z\|_F - \|\mathbf{L}_X\|_F \|\mathbf{S}_X\|_F \|\mathbf{R}_X\|_F. \tag{24}$$

Because $\mathbf{L}_Z$ and $\mathbf{L}_X$ have the same size they are bounded by $\sqrt{n}$ and because $\mathbf{R}_Z, \mathbf{R}_X \in \mathbb{R}^{d \times d}$ and are also normalized, the norm can be bounded by $\|\mathbf{R}_{(\cdot)}\| \leq \sqrt{d}$ respectively. Because we are not in the subspace $\mathbb{R}^s$ but $\mathbb{R}^d$ with $s < d$ and the singular values are $\mathbf{S}_Z, \mathbf{S}_X \in \mathbb{R}^{n \times d}$ the numerical bound becomes

$$D_Z^n = \sqrt{\sum_{i=1}^{n}(p_i^z + |r_i^z|)^2}, \quad D_X^n = \sqrt{\sum_{i=1}^{n}(p_i^x + |r_i^x|)^2}. \tag{25}$$

Applying the bounds to equation 24 we obtain

$$\|\mathbf{L}_Z\|_F \|\mathbf{S}_Z\|_F \|\mathbf{R}_Z\|_F - \|\mathbf{L}_X\|_F \|\mathbf{S}_X\|_F \|\mathbf{R}_X\|_F \leq \|\sqrt{n}D_Z^n\sqrt{d} - \sqrt{n}D_X^n\sqrt{d}\|. \tag{26}$$

Simplifying this term further leads to

$$\|\mathbf{Z} - \mathbf{X}\|_F \leq \sqrt{n}\sqrt{d}\|D_Z^n - D_X^n\|. \tag{27}$$

Because we assume $d > 1$ and a minimum of one non-zero singular vector in $\mathbf{R}_{(\cdot)}$ this at least

$$\sqrt{n}\|D_X^s - D_Z^s\| < \|\mathbf{Z} - \mathbf{X}\|_F \leq \sqrt{n}\sqrt{d}\|D_Z^n - D_X^n\|. \tag{28}$$

$\square$

The result from the above allows the conclusion that the norm of the subspace matrices is smaller than the norm of the original matrices. Note that because BT (Raab & Schleif, 2018) is applied in $\mathbb{R}^d$, this allows the conclusion that $E_{NBT} < E_{BT}$. If the norm of the difference of two matrices is small, they are similar and if they are numerically similar, we can conclude that the distributions are also similar (Song & Park, 2018). Finally, this is the reason why NBT aligns the distribution of two given matrices. However, note that similar distributions not necessarily means a good classification performance in terms of accuracy by an arbitrary classifier in a transfer learning setting.

## A.2    Process of Nyström Basis Transfer

In this section, an illustration of Nyström Basis Transfer (NBT) is given. Figure 2 shows the process of Nyström Basis Transfer. The first row shows the samples of Nyström to create the approximated set of subspace projectors. The second row shows the data after the subspace projection. The similarity in structure but dissimilarity in scaling, as discussed in section 4.3 is visible. The last row shows the data after applying post projection and the difference is not visible anymore.

## A.3    Dataset Description

The study consists of 24 benchmark datasets, already preprocessed and taken from (Gong et al., 2012), (Long et al., 2015) and (Long et al., 2014).

*Reuters*-21578[4] (Long et al., 2015): A collection of Reuters news-wire articles collected in 1987. The text is converted to lower case, words are stemmed and stopwords are removed. With the Document Frequency (DF)-Threshold of 3, the numbers of features are cut down. Finally, Term-Frequency Inverse-Document-Frequency (TFIDF) is applied for feature generation (Dai et al., 2007). The three top categories *organization (orgs)*, *places* and *people* are used in our experiment.
To create a transfer problem, a classifier is not tested with the same categories as it is trained on, i. g. it is trained on some subcategories of organization and people and tested on others. Therefore, six datasets are generated: *orgs vs. places*, *orgs vs. people*, *people vs. places*, *places vs. orgs*, *people vs. places* and *places vs. people*. They are two-class problems with the top categories as the positive and negative class and with subcategories as training and testing examples.

*20-Newsgroup*[5] (Long et al., 2014): The original collection has approximately 20.000 text documents from 20 Newsgroups and is nearly equally distributed in 20 subcategories. The top four categories are *comp*, *rec*, *talk* and *sci* and containing four subcategories each. We follow a data sampling scheme introduced by Long et al. (2015) and generate 216 cross domain datasets based on subcategories: Let $C$ be a top category and $\{C1, C2, C3, C4\} \in C$ are subcategories and analogy $K$ with $\{K1, K2, K3, K4\} \in K$. Select two subcategories each, e. g. $C1$, $C2$, $K1$ and $K2$, train a classifier, select another four and test the model on it. The top categories are respective classes. Following this, 36 samplings per top category-combinations are possible, which are in total 216 dataset samplings. This is summarized as mean over all test runs as *comp vs rec*, *comp vs talk*, *comp vs sci*, *rec vs sci*, *rec vs talk* and *sci vs talk*. This version of *20-Newsgroup* has 25.804 TF-IDF features within 16.021 documents (Long et al., 2015).

*Caltech-256-Office*[6] (Gong et al., 2012): The first, Caltech (*C*) is an extensive dataset of images and contains 30.607 images within 257 categories. The Office dataset is a collection of images drawn from three sources which are from *amazon (A)*, digital SLR camera *(DSLR)* and *webcam (W)*. They vary regarding camera, light situation and size, but ten similar object classes, e. g. computer or printer, are extracted for a classification task. Duplicates are removed, as well as images which have more than 15 similar Scale Invariant Feature Transform (SIFT) in common (Gong et al., 2012).
The final feature extraction is done with Speeded Up Robust Features Extraction (SURF) and encoded with 800-bin histograms. Finally, the twelve sets are designed to be trained and tested against each other by the ten labels (Gong et al., 2012).

Note to reproduce the results of this paper, one should use the linked version of datasets with the same choice of subcategories. A summary of all datasets is shown in table 2. Regardless of the dataset, data have been standardized to standard mean and variance. The experiments are summarized as the mean over 20 runs with a 2-fold sampling, with separated samplings for source and target data.

## A.4    Implementation Details

The parameters for the respective domain adaptation algorithms are obtained via grid-search for best performance on a dataset type, meaning text with Reuters and Newsgroup and image with Office vs Caltech. All algorithms using the SVM with an RBF-Kernel, except GFK which suggest a $k$ nearest

---

[4]http://www.daviddlewis.com/resources/testcollections/reuters21578
[5]http://qwone.com/~jason/20Newsgroups/
[6]https://people.eecs.berkeley.edu/~jhoffman/domainadapt/#datasets_code

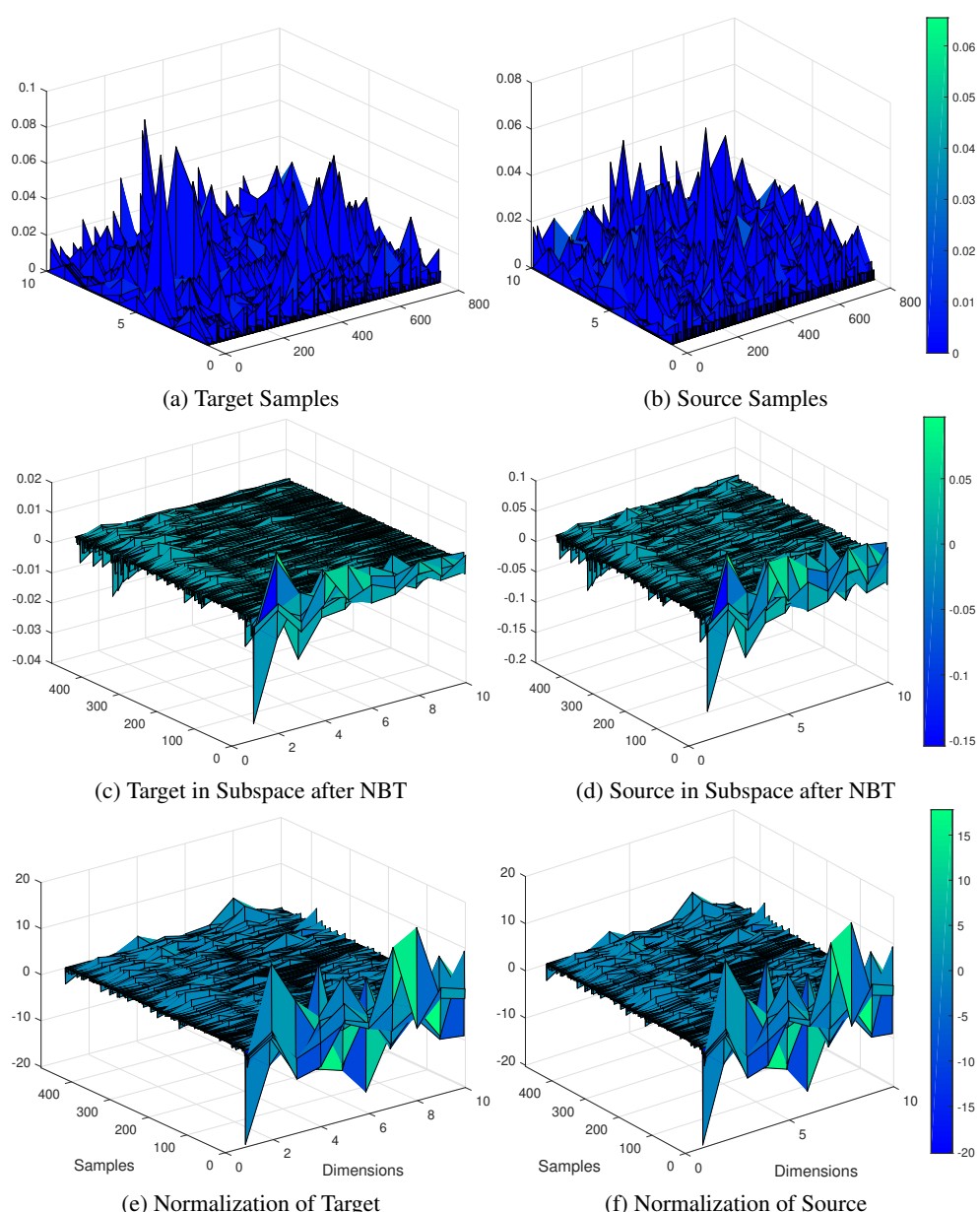

Figure 2: Process of *Nyström Basis Transfer* with ten landmark samples applied to *Caltech vs Amazon* image dataset as a surface plot. The left column shows the target and the right column shows source data. The first row shows samples used for creating NBT transformations. The differences are clearly visible. The second row shows the data after transformation. Note the similarities in structure but differences in scale, based on the approximation error of singular values. The last row contains the normalization correction and differences are hardly visible with the bare eye. This is finally used for training and testing. Note that this is a toy example approximation unsuitable for proper classification due to small landmark size, shown in figure 1. Best viewed in color.

| Office-Caltech Dataset Names | #Samples | #Features | #Labels | Text Datasets Names | #Samples | #Features | #Labels |
|---|---|---|---|---|---|---|---|
| Caltech (C) | 1123 | | | Comp | 4857 | | |
| Amazon(A) | 958 | | | Rec | 3968 | | |
| DSLR (D) | 295 | 800 | 10 | Sci | 3946 | 25804 | 2 |
| Webcam (W) | 157 | | | Talk | 3250 | | |
| | | | | Orgs | 1237 | | |
| | | | | People | 1208 | 4771 | 2 |
| | | | | Places | 1016 | | |

Table 2: Overview of dataset characteristics containing numbers of samples, features and labels. Datasets are categorized into two data types, i.e. text and image. The horizontal line separates the datasets into groups namely Caltech-Office, Newsgroup and Reuters.

| Dataset | SVM Baseline | TCA | JDA | TKL | GFK | SA | CORAL | BT | NBT |
|---|---|---|---|---|---|---|---|---|---|
| Reuters | 0.06 | 0.86 | **0.36** | 0.40 | 3.11 | 0.87 | 8.38 | 0.94 | 0.50 |
| Newsgroup | 1.35 | 21.39 | 4.79 | 2.80 | 214.40 | 59.70 | 705.77 | 34.49 | **2.64** |
| Office-Caltech | 0.05 | 0.64 | 0.45 | 1.09 | 0.38 | 0.28 | 0.37 | 0.20 | **0.09** |
| Overall | 0.48 | 7.51 | 1.77 | 1.21 | 72.54 | 20.22 | 238.38 | 11.87 | **1.08** |

Table 3: Result of experiments shown in mean time in seconds per dataset group. All considered methods using a baseline classifier, hence the SVM is not compared as standalone solution.

neighbor classifier. The $C$ parameter of the SVM is set to 10 for all experiments. For SVM the LibSVM is used. The TCA also has one parameter which gives the subspace dimensions and is set to $\mu = 50$ for both datasets. The JDA has two model parameters. First, the number of subspace bases $k$, which is set to 100. Second, the regularization parameter $\lambda$ is set to 1 for both sets. The TKL approach has the eigenvalue dumping factor $\xi$ as a parameter, which was set to 2 for the text datasets and 1.1 for the image datasets. For the GFK solution, the parameter, number of subspace dimensions is set to 20 for image and 40 for text data. The SA subspace dimension parameter $d$ is set to 5 for every dataset. The CORAL has no free parameters and CGCA is not evaluated, because of numerical issues. The BT approach has no free parameter and for NBT, the subspace dimension is set to 500 for text and 210 for images.

## A.5 TIME COMPARISON

The mean time results in seconds of the cross-validation study per data set group are shown in the table 3. Note that SVM is the underlying classifier for the compared approaches and is presented for the baseline and not listed for stand-alone comparison but included into the time measurement of domain adaptation approaches. In the overall comparison, NBT is the fastest compared to the discussed solutions and in contrast to BT, the required time is a magnitude smaller, especially at Newsgroup, which is a high dimensional dataset. This is an expected effect due to Nyström approximation. Individually, the NBT is the fastest algorithm at the Newsgroup and Office-Caltech data set. At Reuters, JDA is the fastest. In summary, the differences between the winning algorithm and other algorithms are very small. However, GFK and CORAL are the slowest methods with a factor of at least 90 compared to the fastest approach for Newsgroup and are clear outliers in this time comparison.

## A.6 DETAILS OF PREDICTION PERFORMANCE

The detailed version of the results of the experiments are shown in table 4. It is an extended version of table 1 of the main paper.

| Dataset | SVM | TCA 2011 | JDA 2013 | TKL 2015 | GFK 2012 | SA 2013 | CORAL 2016 | BT | NBT |
|---|---|---|---|---|---|---|---|---|---|
| Orgs vs People | 23.6 | 24.0 | 25.3 | 19.5 | 27.8 | 7.0 | 28.8 | 1.2 | 0.2 |
| People vs Orgs | 21.5 | 20.9 | 24.5 | 13.0 | 28.7 | 4.3 | 30.2 | 1.9 | 1.2 |
| Orgs vs Place | 31.9 | 29.0 | 29.5 | 23.5 | 34.7 | 6.5 | 35.6 | 3.4 | 1.7 |
| Place vs Orgs | 34.8 | 32.7 | 34.0 | 18.3 | 35.2 | 7.9 | 37.6 | 4.2 | 2.8 |
| People vs Place | 39.6 | 40.9 | 40.6 | 30.9 | 41.6 | 7.6 | 41.3 | 4.1 | 2.7 |
| Place vs People | 41.2 | 43.0 | 44.6 | 34.1 | 42.1 | 12.5 | 40.6 | 6.0 | 2.7 |
| Reuters Mean | 32.1∗ | 31.8∗ | 33.1∗ | 23.2∗ | 35.0∗ | 7.6 | 35.7∗ | 3.5 | **1.9** |
| Comp vs Rec | 12.7 | 8.1 | 7.8 | 3.0 | 16.3 | 1.8 | 23.0 | 0.3 | 0.7 |
| Comp vs Sci | 24.5 | 26.3 | 27.1 | 9.5 | 24.3 | 4.8 | 28.2 | 0.3 | 0.8 |
| Comp vs Talk | 5.1 | 2.9 | 4.2 | 2.4 | 22.3 | 0.9 | 26.8 | 4.8 | 3.9 |
| Rec vs Sci | 23.7 | 17.3 | 23.9 | 5.1 | 24.4 | 1.6 | 29.5 | 0.1 | **0.2** |
| Rec vs Talk | 18.7 | 13.6 | 15.2 | 5.6 | 23.2 | 1.8 | 29.8 | 5.1 | 3.9 |
| Sci vs Talk | 21.7 | 20.1 | 26.1 | 14.6 | 24.6 | 2.9 | 31.0 | 5.0 | 4.1 |
| Newsgroup Mean | 17.8∗ | 14.7 ∗ | 17.4∗ | 6.7 | 22.5∗ | **2.3** | 28.1 ∗ | 2.6 | **2.3** |
| C vs A | 50.3 | 49.0 | 46.3 | 48.5 | 61.3 | 48.6 | 50.4 | 12.3 | 12.4 |
| C vs W | 58.9 | 59.5 | 55.1 | 53.6 | 63.3 | 81.3 | 60.9 | 20.5 | 20.5 |
| C vs D | 54.3 | 55.7 | 53.4 | 54.2 | 59.7 | 79.2 | 61.8 | 32.8 | 32.8 |
| A vs C | 58.8 | 57.7 | 57.5 | 57.1 | 63.8 | 50.2 | 58.1 | 38.5 | 38.5 |
| A vs W | 66.0 | 63.5 | 57.8 | 58.1 | 65.8 | 73.2 | 60.0 | 20.5 | 20.5 |
| A vs D | 59.2 | 61.5 | 62.6 | 56.8 | 64.5 | 74.2 | 61.8 | 34.3 | 34.3 |
| D vs C | 76.8 | 71.3 | 68.7 | 65.5 | 73.4 | 78.1 | 67.4 | 36.3 | 36.3 |
| D vs A | 72.1 | 67.8 | 66.5 | 64.0 | 68.4 | 70.5 | 64.5 | 4.7 | 4.8 |
| D vs W | 24.4 | 23.8 | 18.9 | 23.6 | 22.3 | 45.4 | 18.9 | 32.8 | 32.8 |
| W vs C | 74.4 | 68.6 | 68.0 | 64.4 | 70.8 | 80.4 | 68.3 | 33.4 | 33.5 |
| W vs A | 75.2 | 65.3 | 66.4 | 63.6 | 69.7 | 73.7 | 66.3 | 6.8 | 6.8 |
| W vs D | 52.9 | 31.7 | 27.7 | 23.5 | 33.1 | 33.2 | 25.8 | 19.8 | 19.8 |
| Office-Caltech Mean | 55.5 ∗ | 51.5 ∗ | 51.4 ∗ | 49.2 ∗ | 55.3 ∗ | 68.7 ∗ | 52.6 ∗ | **20.7** | 20.8 |
| Overall Mean | 35.1 ∗ | 32.7 ∗ | 34.0 ∗ | 26.4 ∗ | 37.6 ∗ | 26.2∗ | 38.8 ∗ | 8.9 | **8.3** |

Table 4: Result of cross-validation test shown in mean error in percent per dataset. Mean over dataset group at the end of each section. Bold marks winner. ∗ marks statistical differences with a p-value of 1% against NBT. The study shows that none of the listed algorithms is statistically significant better as NBT.

