# OpenReview forum: "Domain Adaptation via Low-Rank Basis Approximation"
_ICLR.cc/2020/Conference — Reject_

### Official Review · AnonReviewer2 · 2019-10-21
**Official Blind Review #2**

**Rating:** 1

**Review:**

This paper is mainly an improvment of the basis transfer model, by introducing the Nystrom approximation and a NBT model is formulated. This paper lacks of research motivation and solid experimental validation.
The authors claim "it is the fastest domain adaptation algorithm in terms of computational complexity", which is not very convinced.
I cannot observe new knowledge in domain adaptation, except the Nystrom approximation technique used.
The equation (9) is similar to the representational based transfer model, where the T matrix is just the reconstruction matrix, because X and Z have different number of samples.
This paper written is a little poor, and the novelty of NBP is not clearly claimed.
The experimental effectiveness is weak and have no improvement compared with BT 2018 in image dataset.
In deep learning era, could you discuss what is the value in deep transfer learning or deep domain adapatation?

**Experience Assessment:**

I have published in this field for several years.

**Review Assessment: Checking Correctness Of Derivations And Theory:**

I assessed the sensibility of the derivations and theory.

**Review Assessment: Checking Correctness Of Experiments:**

I carefully checked the experiments.

**Review Assessment: Thoroughness In Paper Reading:**

I read the paper at least twice and used my best judgement in assessing the paper.

---

> ### Author Response · Authors · 2019-11-11
> **Comment to Review 2**
>
> Thanks for the review. Please find our statements subsequently to the various points mentioned in your review. We do not expect that this comment will substantially
> change your opinion but believe that some clarifications are required which should
> be in your interest and also in the interest of the area chair or respectively the
> honored organizers of this ICLR 2020 conference.
>
>
> "This paper lacks research motivation and solid experimental validation."
> > As mentioned in the paper, the motivation is discussed in section 1. And the benefits and differences to related work are further considered in section 2, section 4, and section 5.
> > For the reasoning about experimental design, see the comments to review 1 and 2.
>
> 'The authors claim "it is the fastest domain adaptation algorithm in terms of computational complexity" which is not very convinced.'
> >Honestly, we do not know how to respond to this. We showed in landau notation and experimental results that the method is efficient.
>
> "I cannot observe new knowledge in domain adaptation, except the Nystrom approximation technique used. "
> >As mentioned in the paper, the override of the basis of source in the target subspace leads to effective domain adaptation. Note the proposed approach is detailed at the beginning of eq. 11.
>
> "The equation (9) is similar to the representational based transfer model, where the T matrix is just the reconstruction matrix because X and Z have a different number of samples."
> >As mentioned in the paper, the NBT does -- not -- need a reconstruction matrix and also eq. 9 is -- not -- the NBT formulation but a recap of related work.
> -- Please consider the correct approach beginning at eq. 11. --
>
> This paper written is a little poor, and the novelty of NBP is not clearly claimed.
> >We claimed it four times in section1, section 2, section 4, and section 5. We clearly want to make sure the reader gets it.
> >We cannot extract useful feedback given "a little poor" please provide constructive feedback.
>
> "In deep learning era, could you discuss what is the value in deep transfer learning or deep domain adaptation?"
> > This paper was not at all-around deep learning - on purpose - and due to the aforementioned reasons. Please see the comment to reviewer 1.
>
>
> Thank you for reading.

---

### Official Review · AnonReviewer3 · 2019-10-23
**Official Blind Review #3**

**Rating:** 3

**Review:**

1.	This manuscript is mainly based on the previous work BT (Raab and Schleif, 2018). The novelty seems to be too limited.
2.	The proposed method NBT improves the computional efficiency of BT. The Datasets used in the experiments are not representative. More experiments should be conducted to demonstrate its efficiency on large-scale Datasets, such as VisDA and Offic-Home Dataset.
3.	The precise results of CGCA are not provided, which is unfair. Besides, there is no comparsion between the proposed methods and the state-of-the-art deep learning-based methods.The experimental results seems unconvincing.
4.	Typos:
(1)	‘Out BT approach has no free parameter…’ in page 14. Here ‘Out’ means ‘Our’?
(2)	‘…NBT is fastest at Newsgroup und Image data set. In page 14.’


**Experience Assessment:**

I have read many papers in this area.

**Review Assessment: Checking Correctness Of Derivations And Theory:**

I assessed the sensibility of the derivations and theory.

**Review Assessment: Checking Correctness Of Experiments:**

I assessed the sensibility of the experiments.

**Review Assessment: Thoroughness In Paper Reading:**

I read the paper at least twice and used my best judgement in assessing the paper.

---

> ### Author Response · Authors · 2019-11-11
> **Comment to Review 3**
>
> Thanks for the review. Please find our statements subsequently to the various points mentioned in your review. We do not expect that this comment will substantially
> change your opinion but believe that some clarifications are required which should
> be in your interest and also in the interest of the area chair or respectively the
> honored organizers of this ICLR 2020 conference.
>
> 1. "This manuscript is mainly based on the previous work BT (Raab and Schleif, 2018). The novelty seems to be too limited."
> >This sentence contains no useful feedback. The only thing we have in common with BT is the principle. We discuss this in various points throughout the paper.
>
> 2. The Datasets used in the experiments are not representative.
> >The proposed datasets are commonly used in the area of domain adaptation and recently used.
> Further, we showed in Landau notation that we have lower complexity. Therefore it is representative.
>
>
> 3. CGCA is not provided
> >For CGCA a comparison is complicated due to numerical errors in the - original - implementation. We have provided the obtained results in the experiments.
> The deep adaptation techniques are missing, mainly because we proposed a non-deep learning approach. As you can easily check, we have evaluated our approach on the variety of typically
> used benchmark data. There are additional very good reasons not to focus on deep learning approaches alone. Deep learning requires a huge amount of reliable (labeled) training data which is not applicable in many real life problems (e.g. manufacturing domains, for small to medium scale companies a.s.o). In those settings, we may have rather a few 100 to thousands of data but not at the scale of deep learning. This is a particular motivation why non-deep learning methods are also extended to transfer learning. Otherwise, we could completely abandon everything aside of deep learning. This paper was not around deep learning - on purpose - and due to the aforementioned reasons.
>
>
> 4. " (1) 'Out BT approach has no free parameter…' on page 14."
> >Sorry, but this sentence does not appear on page 14. Not even in the whole paper.
>
> 4.2 Thank you, we corrected it.
>
> Thank you for reading.

---

> > ### Comment · AnonReviewer3 · 2019-11-14
> > **Re**
> >
> > > Authors claim that the proposed method is the fastest domain adaptation algorithm in terms of computational complexity.  It is necessary to demonstrates this statement experimentally, especially in large-scale datasets.
> >
> > > For fairness,  authors can not compare with an algorithm that has no definite or correct results.
> >
> > > "Out BT approach has no free parameter, and for NBT, the subspace dimension is set  to 500 for text and 210 for images."  (on page 14, the last sentence in the subsection of A.4).

---

> > > ### Author Response · Authors · 2019-11-15
> > > **Re**
> > >
> > > > Authors claim that the proposed method is the fastest domain adaptation algorithm in terms of computational complexity.  It is necessary to demonstrates this statement experimentally, especially in large-scale datasets.
> > > We proposed a domain adaptation algorithm. Hence, we have tested it on domain adaptation datasets. Notably, the Newsgroup is the highest dimensionally, which is crucial for evaluating computational time. But also the dataset has a high number of samples compared to other DA datasets.
> > > Further, we claimed not to be the fastest, but the fastest in comparison to the tested algorithms.
> > >
> > > > For fairness,  authors can not compare with an algorithm that has no definite or correct results.
> > > We removed CGCA out of the experiments to be aligned with the review.
> > >
> > > > "Out BT approach has no free parameter, and for NBT, the subspace dimension is set  to 500 for text and 210 for images."  (on page 14, the last sentence in the subsection of A.4).
> > > This is not an "Out", but a "The" and has no spelling issue.

---

### Official Review · AnonReviewer1 · 2019-10-23
**Official Blind Review #1**

**Rating:** 1

**Review:**

In this work, the authors improve the work (Raab & Schleif, 2018) by (1) reducing the computational complexity, (2) neglecting the sample size requirement, and (3) achieving a low-rank projection through Nystrom approximation. More specifically, the feature dimensionality is reduced by using only s biggest eigenvalues and eigenvectors, and the sample size is coordinated through Nystrom approximation. Class-wise sampling is used for the source, and uniform sampling is used for the target. Experimental studies on three datasets have been done.

This paper should be rejected because (1) the paper lacks important latest references on domain adaptation, (2) the paper misuses the notations that makes the paper is not easy to follow, (3) the algorithm is not well justified either by theory or experiments, and (4) the presentation should be further polished. Here are some detailed comments:

(1)	A lot of recent deep domain adaptation methods are missing. These deep works achieve the state-of-the-art results on many transfer benchmark tasks. Without comparison with them (the paper does not mention any deep works), it is very unconvincing to conclude the paper makes new contributions to the transfer community.
(2)	In line 2 of page 5, the authors claim that BT assumes S_Z ~ S_X, which is not true. The key idea of BT is to construct new source data using target basis and source eigenvalues. Similarly, the authors make the same claim in section 4.1 for NBT, which is also not valid.
(3)	The notations in eq. (12) and (13) are very misleading. Eq. (12) follows the notations of the first line in page 5, but in the first line below eq. (12), why R_X \in R^{d \times s}, S_X^2 \in R^{s \times s}? I understand that X_s is the low dimensional X, then X_s = XR_s with R_s is the dimensionality reduction matrix whose size is d \times s (using biggest s eigenvectors is fine). With this, eq. (13) is incorrect as X should be L_X*S_X*R_X^T, but not L_s*S_s*R_s^T. Moreover, it is also unclear why X is decomposed into product of two matrices in this work, is there any benefit of doing so for transfer purpose?
(4)	In section 4.1, A_Z and A_X have exactly same form with X and Z, i.e., L_Z*S_Z*R_Z^T and L_X*S_X*R_X^T, please clarify. How eq. (14) X_s = \tide{L_X}*S_X come from? Is it the same as eq. (13)?
(5)	The title highlights low-rank, but it is not very clear how low-rank matters in the proposed method. I do not find contents stating the low-rank property of the proposed algorithm in the main technical sections.
(6)	Regarding section 4.2, what is the benefit of using class-wise sampling for the source? Have you tried to use uniform sampling for both the source and target domains?
(7)	The bound in eq.(16) is not very meaningful as s << n, m, d. Moreover, I am also not convinced by the claim this reduces the distribution differences, please give more theoretical justifications.
(8)	Regarding the experimental studies, why not use accuracy as many existing works do? The baselines are all subspace-based papers, and are out-of-the-date. The latest subspace papers, e.g., JGSA and MEDA, should be included. Moreover, deep methods are completely missing, which makes the empirical evaluation much less convincing. The improvements of NBT to BT are very marginal, 0.6.
(9)	Some typos and unclear points (please further polish the paper):
(a)	The last sentence of para 2, X and Z should be data, not features/
(b)	Para 3, it is unclear what are the implicit alignment and explicit alignment of domain distributions.
(c)	The first sentence in section 2, it should be homogeneous.
(d)	Page 5 above eq. (15), it should be S_Z ~ S_X.
(e)	Section 4.2 the second line, it should be “in the data matrix”
(f)	The second last line of page 6, it should be inequality (16).
(g)	Some references miss page information.



**Experience Assessment:**

I have published in this field for several years.

**Review Assessment: Checking Correctness Of Derivations And Theory:**

I assessed the sensibility of the derivations and theory.

**Review Assessment: Checking Correctness Of Experiments:**

I carefully checked the experiments.

**Review Assessment: Thoroughness In Paper Reading:**

I read the paper at least twice and used my best judgement in assessing the paper.

---

> ### Author Response · Authors · 2019-11-11
> **Comment to Review 1**
>
> Thanks for the review. Please find our statements subsequently to the various points mentioned in your review. We do not expect that this comment will substantially
> change your opinion but believe that some clarifications are required which should
> be in your interest and also in the interest of the area chair or respectively the
> honored organizers of this ICLR 2020 conference.
>
> >>  This paper should be rejected because [...]
>
> (1) the paper lacks important latest references on domain adaptation,
> >In your opinion, we miss just two papers: MEDA and JGSA. The rest of the related work is partly very recent.
> We define related work as related to our work, which in this case are subspace approaches
> for transfer learning in traditional learning models, where only a moderate amount of data may be available. Deep learning approaches for transfer learning are competitive approaches
> to the considered methods but are not in particular focus of our approach nor the other
> very recent high ranked work we compared within the paper.
> The methods MEDA and JGSA have been published -- very -- recently.
>
> (2) the paper misuses the notations that makes the paper is not easy to follow,
> >We are using a standard domain adaptation notation. But for two domain matrices with singular value decompositions with their low-rank versions, we got 14 different matrices. We are sorry that it requires a variety of notations.
> You claim that "In line 2 of page 5, the authors claim that BT assumes S_Z ~ S_X, which is not true."
> > This is mentioned in the paper, and we invalidated this by ourselves for some cases. Take a look at the pseudo-code and figure 2. How can we get a reject for something we claim by ourselves and provide a possible solution for it?
> Further, the critiques on wrong equations or claims (3,4,6,8) are very detailed before or after the equations, respectively. Finally, we call it low-rank approach because we are using approximated low-rank matrices via Nyström.
>
>
> (3.1) the algorithm is not well justified either by theory
> >The theory of this approach is based on the singular value decomposition, eigenvalue decomposition, and the orthogonal Procrustes problem. These techniques and problems are well known and studied for decades. We do not change the mechanics of the approaches but interpreted the results for domain adaptation techniques, and therefore, the theory is also valid. Your current complains about the theory in the paper are too vague to help us in improving the paper. You mentioned that the bound is not very meaningful but actually shows that matrices are very similar after NBT resulting in similar distributions and is perfectly fine for our case.
>
> (3.2) or experiments
> >The deep adaptation techniques are missing, mainly because we proposed a non-deep learning approach. As you can easily check, we have evaluated our approach on the variety of typically
> used benchmark data. There are additional very good reasons not to focus on deep learning approaches alone. Deep learning requires a huge amount of reliable (labeled) training data which is not applicable in many real life problems (e.g. manufacturing domains, for small to medium scale companies a.s.o). In those settings, we may have rather a few 100 to thousands of data but not at the scale of deep learning. This is a particular motivation why non-deep learning methods are also extended to transfer learning. Otherwise, we could completely abandon everything aside of deep learning. This paper was not around deep learning - on purpose - and due to the aforementioned reasons.
> We will integrate MEDA into the revision of the paper, but this will not change the overall picture. We did not do any cherry-picking.
>
> >The reviewer misses the essential part of this paper. We do not learn or find a new projections matrix. We interpret the solutions of the orthogonal Procrustes problem to give a subspace transformation. This paper shows that for effective domain adaptation, the learning of complicated functions is not necessarily required.
>
>
> >Note, we subsequently will provide the experimental results for MEDA, which seems to be an important requirement for acceptance. Parts are already included right now.
>
> (4) Thank you for the feedback. We fixed the issues mentioned in point 9.
>
>
> Thank you for reading.

---

### Decision · Program_Chairs · 2019-12-19

**Decision:**

Reject

**Comment:**

Three reviewers have scored this paper  as 1/1/3 and they have not increased their rating after the rebuttal and the paper revision. The main criticism revolves around the choice of datasets, missing comparisons with the existing methods, complexity and practical demonstration of speed. Other concerns touch upon a loose bound and a weak motivation regarding the low-rank mechanism in connection to DA. On balance, the authors resolved some issues in the revised manuscripts but reviewers remain unconvinced about plenty other aspects, thus this paper cannot be accepted to ICLR2020.